# Translation, Cross-Cultural Adaptation, and Feasibility of the NHPT-E of Manual Dexterity for the Spanish Population

**DOI:** 10.3390/healthcare12050550

**Published:** 2024-02-27

**Authors:** Gema Moreno-Morente, Miriam Hurtado-Pomares, Alicia Sánchez-Pérez, M. Carmen Terol-Cantero

**Affiliations:** 1Department of Pathology and Surgery, Universidad Miguel Hernández de Elche, 03550 Alicante, Spain; mhurtado@umh.es (M.H.-P.); alicia.sanchez@umh.es (A.S.-P.); 2Grupo de Investigación en Terapia Ocupacional (InTeO), Universidad Miguel Hernández de Elche, 03550 Alicante, Spain; 3Instituto de Investigación Sanitaria y Biomédica de Alicante (ISABIAL), 03550 Alicante, Spain; 4B + D + b Occupational Research Group, Miguel Hernández University of Elche, 03550 Alicante, Spain; 5Department of Behavior and Health, Miguel Hernández University of Elche, 03550 Alicante, Spain; macarmen@umh.es; 6Research Group—Psychosocial Action in the Community Sphere, 03550 Alicante, Spain

**Keywords:** adaptation, assessment, feasibility, manual dexterity, nine-hole peg test, Spanish

## Abstract

The Nine-Hole Peg Test (NHPT) is considered a “gold standard” for the measurement of manual dexterity. The aim of this study was the translation and culturally adapting the original version of the NHPT. Materials and Methods: The adaptation was carried out following the standardized translation–retrotranslation guidelines and procedures referred to in the literature and in the International Test Commission (CIT). The final Spanish version of the NHPT (NHPT-E) was administered to 40 healthy adults. We evaluated its feasibility by means of a questionnaire elaborated according to Iraossi’s checklist proposal for the pilot test process. Results: Modifications of expression in the grammatical mode of the verbs were performed, as well as the adaptation of some terms used in the three sections of the original version of the test (General Information, Installation, and Application Instructions). In the pilot study, for 95% of the participants, the NHPT-E is a comfortable test to take, and, for 100% of the evaluators, the test includes all the necessary information, with clear instructions and interpretation of the results. Conclusions: The cross-cultural adaptation and pilot study enabled the development of a suitable and viable version of the NHPT-E for use in the Spanish population.

## 1. Introduction

Dexterity is the ability of a person to use their fingers, hands, and arms to perform tasks such as activities of daily living, work, school, play, and leisure [1,2]. Manual dexterity can be impaired by traumatic injuries to the hands or by certain rheumatologic or neurologic pathologies (arthritis, osteoarthritis, stroke, multiple sclerosis, and Parkinson’s disease, among others). These lesions cause limitations of movement [3,4] in the hand and manual function, with significant negative consequences on the performance of activities of daily living [5], on the ability to live independently, and on quality of life (QoL) [6,7]. These functional and psychosocial limitations can even cause loss of work and the need for specific care that entails high economic and social costs [8]. Considering the impact caused by manual dexterity impairments, correct evaluation of hand and arm function is essential as part of a comprehensive assessment in the process of therapeutic effectiveness and development of treatment strategies [9].

There are different standardized tests [10,11,12] that evaluate aspects of fine motor skills, such as speed and precision of movement, grasp and release, writing skills, and hand posture. In general, these tests are short, easy to apply, objective, and do not require expensive previous training. One of the most widely used and recommended by experts to assess manual dexterity is the Nine-Hole Peg Test (NHPT) [1,13,14,15], a brief and easy-to-use test originally designed in 1971 by Kellor et al. [13] and considered the “gold standard” for the measurement of manual dexterity. Subsequently, Mathiowetz et al., in 1985 [14], detailed the instructions for its administration [14], analyzed its psychometric properties of reliability and validity, and presented normative data [14,15]. The NHPT has been widely used in different populations, mostly with neurological pathologies such as Parkinson’s disease [16], multiple sclerosis [17], and stroke [18], among others. Validation studies carried out with neurological disease show that this tool presents optimal psychometric properties, with high reliability indices in terms of an interclass correlation coefficient (ICC) between 0.95 and 0.99 [15,19,20] and validity analyses correlating NHPT with hand grip and finger pinch. In addition, NHPT has been administered in countries such as the United States in healthy populations of children [2,21,22] and adults [15], proving to be an effective tool for screening manual dexterity in both population types [21,23]. 

In Spain, both in clinical practice and in the field of research, its use is increasing in the field of neurology and rehabilitation. Although in Spain the NHPT has been included in several studies of the adult population with neurological pathology [24,25], no data on the cross-cultural adaptation process have been reported. Only one study that focused on a Spanish sample of adults with cerebral palsy (CP) [19] reports data on the reliability of the NHPT in a population of 27 patients with unilateral spastic CP (ICC = 0.94 on the affected side and ICC = 0.96 for the unaffected side). In view of the above, the aim of this study was to carry out a cross-cultural adaptation of the NHPT and to test its feasibility for administration in the Spanish population by means of a pilot study. 

## 2. Materials and Methods

### 2.1. Process of Transcultural Adaptation of the NHPT to Spanish (NHPT-E)

The material subject to cross-cultural adaptation of the NHPT includes the three sections of the original version [14] (General Information, Set-up, and Patient Instructions) plus the response log. 

To carry out the present study, permission was obtained from the original author (Mathiowetz et al., 1985 [14]) by e-mail request. The adaptation was carried out following the standardized translation–retrotranslation guidelines and procedures referred to in the literature and the International Test Commission (CIT) (Figure 1) [26,27]. The phases of the procedure were as follows: (1) two bilingual expert translators who did not know the original version performed the direct translation independently from English to Spanish (T1 and T2); (2) a committee of experts formed by nine occupational therapists, applying the principles of equivalence, compared and synthesized the two translations (T1 and T2) and presented the first Spanish version of the NHPT (NHPT-E1). The committee of experts was selected for their experience in the use of manual dexterity evaluation instruments as well as in carrying out research studies on the translation and cross-cultural adaptation of health measurement tools from Miguel Hernández University of Elche (Spain); (3) another bilingual translator, also unfamiliar with the original version of the NHPT, back-translated the NHPT-E1 (T3) into English; (4) the expert committee reached a consensus on any discrepancies among all documents (original questionnaire, T1,T2, NHPT-E1, and T3) and presented the version NHPT-E2; and (5) in the pilot study, the NHPT-E2 was administered to test the feasibility. 

### 2.2. Pilot Testing of the NHPT-E2

#### 2.2.1. Participants

Forty healthy adults from different municipalities of the Valencian Community, selected by convenience, participated in the study. Sixty percent were women, with a mean age of 39 years, and 33% had primary education. The sociodemographic characteristics of age, gender, educational level, and employment status of the participants (collected by means of an ad hoc questionnaire) are presented in Table 1. The inclusion criteria were being over 18 years of age, not presenting visual and/or auditory perceptual, language, upper limb mobility, and/or comprehension alterations that limited the performance of the tests, not presenting another systemic disease associated with cognitive impairment or a history of severe psychiatric disease (depression, psychosis, or schizophrenia), and/or alcohol and drug abuse. All participants previously signed an informed consent form. The study was approved by the ethics committee of the Universidad Miguel Hernández (DPC.GMM.01.20).

#### 2.2.2. Instrument


-NHPT-E2: It consists, like the original version (NHPT), of a rectangular base composed of a small container with nine holes (10 mm diameter and 15 mm deep) and nine pins (7 mm diameter and 32 mm long). The recommended commercial plastic version of the NHPT [28] was used after finding that there is no significant difference in the time to complete the task between this version and the wooden version [14]. To perform the test, the person being tested must insert the nine pegs, one at a time, into the holes as quickly as possible. Subsequently, the pegs must be removed, one at a time, and placed back into the container. This process must be completed with both hands independently, and the test is always started with the dominant hand, recording the time (in seconds) it takes the person being tested to complete the task with each hand.-Feasibility: an evaluation questionnaire developed from Iraossi’s pilot test process checklist was used [29]. It included 5 questions for participants focusing on (1) the purpose of the test, (2) comfort in taking the test, and (3) clarity of the prompts. Another 14 questions were intended for the evaluators (*n* = 4), related to the aforementioned information (1,2,3), and also to the ease or difficulty in its application, recording, and interpretation of results. The answers for the total of the 19 questions were Likert-type (not at all; I don’t think so; I am not sure; I think so; and yes totally).


#### 2.2.3. Statistical Analysis

The free statistical software R. 4.2.0 was used (https://www.r-project.org/) (accessed on 4 July 2023). Descriptive analyses were performed using frequencies and percentages for categorical variables, and mean and standard deviation, as well as median and interquartile range, were used to describe quantitative variables. 

## 3. Results

### 3.1. NHPT-E2 Adaptation

In phases (1) and (2) of the adaptation process, the committee of experts carried out a process of equivalence analysis and comparative review of the translations of the original version (T1 and T2), which resulted in the first version of the Spanish NHPT (NHPT-E1). In this process, the following changes were proposed: (1) for grammatical expressions of verbs, it was chosen to use verbal periphrases of the infinitive “debe + infinitive” instead of “debe de + infinitive” since the latter is used when one wants to express an assumption or a belief; (2) for expressions such as “paciente”, “clavijero”, “test”, “testar”, and “esto es una práctica”, it was decided to use “persona evaluada”, “tablero”, “prueba”, “evaluar”, and “esto es una prueba”, respectively, as they are considered to be more frequently used in Spanish; (3) for the measurement system of the instrument, the use of centimeters was agreed upon; (4) the name of the section “Instructions to the patient” was changed to “Instructions”; and (5) the following information was moved from the “Installation” section to the “Instructions” section: “The board should be placed in front of the subject being evaluated, with the container containing the pegs on the side of the dominant hand” (Table 2).

Finally, after the back-translation process (T3) of phase 3, the committee of experts compared it again with the original version (phase 4) and decided to modify the expression “Installation” to “Construction measures”, giving rise to the second version of the Spanish NHPT (NHPT-E2) used in the pilot study. Table 2 shows the final result of this process.

### 3.2. NHPT-E2 Pilot Study

Phase 5 of the NHPT cultural adaptation process corresponds to the results obtained from the pilot study. 

Table 3 presents the NHPT-E2 scores stratified by sex, employment status, and educational level. The time taken to perform the test with the dominant hand was greater in women, in people with secondary education, and in the unemployed. On the other hand, with the non-dominant hand, the time taken to perform the test was also greater in women, in people with primary and secondary education, and in active people.

### 3.3. Feasibility of NHPT-E2

In the case of the responses of the person evaluated and during the pilot study to the questions about the purpose of the test, 55% and 87.5% responded that they knew what it was for and that manual dexterity was what the researchers wanted to know about when administering it. In addition, 95% of the participants acknowledged that they felt comfortable performing it, and, for 100%, the test was not long at all. The total number of participants evaluated reported that the indications for performing the test were clear, that they had no difficulty in knowing what they had to do, nor was there any confusion with the indications for performing the test.

With regard to the evaluators’ assessment of the purpose of the test, 100% stated that the test gathered the necessary information to determine the manual dexterity of the person being evaluated, as well as that it was simple, short, and easy to perform. They also agreed that the structure and wording of the instructions were equally understandable. As for the assessment of the application, recording, and interpretation, 100% also agreed that it was not difficult or complex and that the results were easily interpreted. 

After the pilot study, in a final review of the expert committee with the evaluators, it was agreed to modify the instruction on the removal of the pins of the NHPT-E1 and NHPT-E2 versions from “Now take them out … faster” to “Now take them out one at a time … faster”. With this modification, the final version of the NHPT-E was determined (Appendix A), which consists of three sections, as in the original version of the test, and the answer sheet where the total score of the test, measured in seconds, is recorded. 

## 4. Discussion

In this study, a process of cross-cultural adaptation of the NHPT to Spanish has been carried out, in addition to demonstrating its feasibility with the results obtained in the pi-lot test of the pre-final version of the NHPT-E. Firstly, and with respect to the process of cultural adaptation, it should be said that the NHPT has been used in numerous countries, but, for the most part, there is no information in the scientific literature on the prior process of cultural adaptation, a process that ensures its adequate applicability, above all, in the case of clinical evaluations [14,26,27]. Although in all these studies the NHPT administration procedure was carried out according to the original instructions established by Mathiowetz et al. [14], this work has focused on the need to carry out these translation and back-translation procedures, which, together with the review by the expert committee, show the changes required to culturally adapt the original version. Thus, the results of the study show that the final version of the NHPT-E maintains the three sections of the original version, as well as the number of items, but the grammatical expressions of the verbs have been modified, in addition to adapting some terms that are more culturally and idiomatically appropriate in the Spanish population [32]. 

Second, during the pilot study of the NHPT-E, as recommended by the original study, it was administered using an anti-slip mechanism under the board [14]. The results of the pilot study indicated that participants between the ages of 20 and 39 years, men, persons with higher education, and active persons took less time to complete the test, both with the dominant and non-dominant hand. The time taken to complete the test according to age is in agreement with previous studies, showing a certain deterioration of manual dexterity in older people, which is understood to be part of a healthy aging process. Similarly, in the case of manual dexterity according to sex, and taking into account the median value, women and men show different scores (15.42 and 15.05, respectively), which, coinciding with other studies, could be explained as a function of lifestyle, or the greater or lesser involvement in certain activities that involve fine manipulation of certain everyday objects [33]. However, a much larger sample would be needed to evaluate the properties of the measure in various age and sex subgroups.

Finally, regarding the feasibility results, in general, the participants did not report any difficulties or relevant problems with the NHPT-E version. They all considered that the instructions were clear and perfectly understood. In addition, they considered that they understood the purpose of the test and that they found it easy and brief to complete. For their part, the evaluators’ references about the NHPT-E were that it collected the necessary information to measure manual dexterity, which would support its apparent validity, confirming that the instrument measures what it intends to [34]. In addition, it is the evaluators who also pointed out the ease of its application, recording, and interpretation of results, agreeing, finally, that it is a simple, brief, and convenient test for use in clinical evaluations. These references and feasibility results assure us that the NHPT-E conforms to the principles of “parsimony” to be considered in the case of administration and correction of questionnaires or evaluation tests in clinical contexts [35]. 

### Limitations

The present study has some limitations that should be considered. The findings are based on a small sample of participants from a geographic area in Spain, but it should be noted that the pilot study used an adequate sample size according to the recommendations of Beaton et al. [27], and that the translation process was adjusted to the use of standard Spanish, which is widely understood by the entire Spanish-speaking population. On the other hand, regarding the number of trials performed for familiarization with the test and to determine whether the person has understood the test instructions, and since there is no consensus among researchers [19], one was performed with each hand and not three, as recommended by Grice et al. [15], in their study performed with the commercial version of the test. We believe that a single practice test was sufficient and no learning effect was produced. Furthermore, Feys et al. [19] question the repeated practice of the test as this may imply better training of the test takers and therefore better scores. In turn, this would imply a longer time for test administration and perhaps greater fatigue of the test takers. However, taking into account the suggestions of the original study, a commercial plastic version from Smith and Nephew [15] with a round container [14] was used that conforms in terms of dimensions to the original proposal of Mathiowetz et al. [14]. This allowed us to reduce the need for pre-test trials by reducing the difficulty for the test subjects to pick up the pegs of the wooden container in the original study [14].

## 5. Conclusions

Therefore, by way of conclusion, this study demonstrates that the adapted version of the NHPT for the Spanish population, the NHPT-E, is presented as an instrument that is (1) simple, clear, and easy to understand, in accordance with the proposal of the original version; (2) it maintains the same response format; and (3) the changes made have improved the understanding of the test for participants and the administration and correction process for evaluators.

### Clinical Applicability and Future Research

Having demonstrated the feasibility of the final version translated and adapted for the Spanish population, the NHPT-E is shown to be a useful test that will facilitate evaluation in clinical contexts, and it ensures homogenization for its administration and correction among evaluators. This will allow the possibility of comparison across different population groups in successive studies and investigations, as well as ensuring the contrast of results between examiners, which will corroborate its internal validity. The NHPT-E may also be the clinical and diagnostic test of choice for neurological pathologies at different stages of the disease process. In future research, as well as in the follow-up of interventions or treatments with different neurological samples, its use will be consolidated as a “gold standard” tool for measuring manual dexterity, endorsing the references of numerous experts in the field of rehabilitation regarding this instrument. Similarly, the NHPT-E is a useful assessment test for healthy populations, comparing the development and deterioration of manual dexterity throughout the life cycle, urging us to continue the process of research and study of its psychometric properties in different representative samples of the Spanish population, including men and women in different age ranges.

## Figures and Tables

**Figure 1 healthcare-12-00550-f001:**
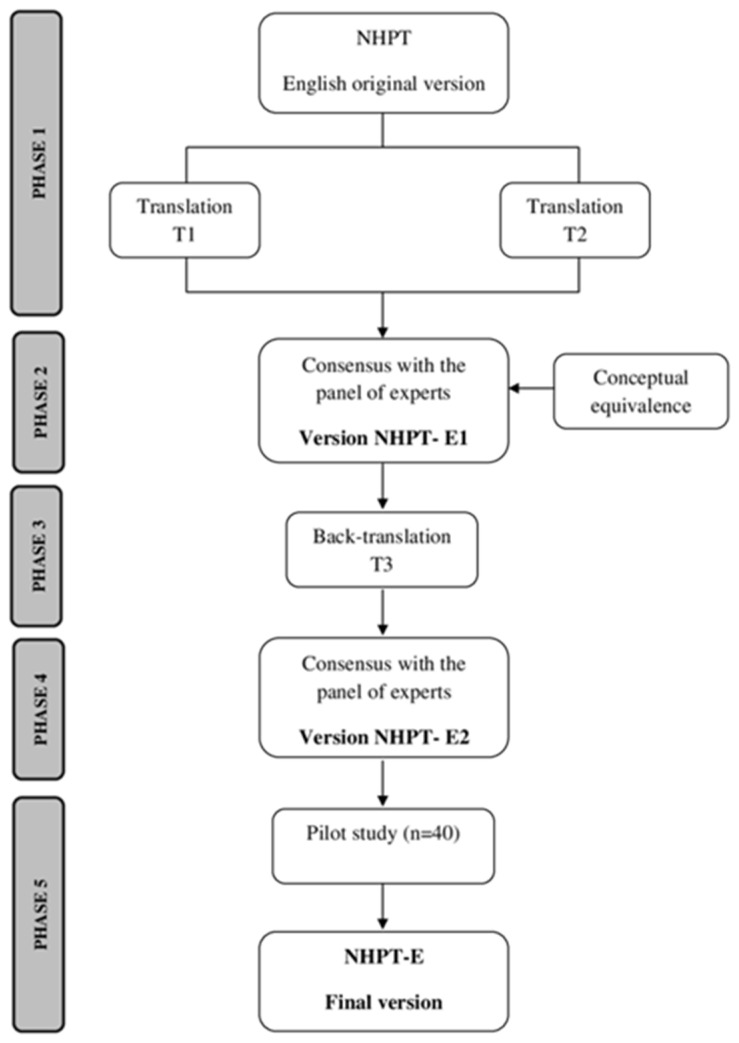
Cultural adaptation process of the NHPT tool.

**Table 1 healthcare-12-00550-t001:** Sociodemographic characteristics of study participants (*n* = 40).

Age, Mean (IR)	39 (21;59)
Gender, *n* (%)	
Female	24 (60)
Male	16 (40)
Educational level, *n* (%)	
No or primary education	13 (32.5)
Secondary	11 (27.5)
Superiors	16 (40)
Employment status, *n* (%)	
Active	35 (87.5)
Unemployed	5 (12.5)
Dominance, *n* (%)	
Right-handed	37 (92.5)
Lefty	3 (7.5)

*n*: number of participants; IR: interquartile range.

**Table 2 healthcare-12-00550-t002:** Results of the equivalence analysis process.

	Mathiowetz et al., 1985 [14]	Committee of Experts
**Información** **Ormación General**	- The Nine-Hole Peg Test should be conducted with the dominant arm first.	- La Prueba del Tablero de Nueve Agujeros debe realizarse primero con el brazo dominante.
	- One practice trial (per arm) should be provided prior to timing the test.	- Debe hacerse un ensayo antes de cronometrar la prueba con cada brazo.
	- Timing should be performed with a stopwatch and recorded in seconds.	- El tiempo debe medirse con un cronómetro y registrarse en segundos.
	- The stop watch is started when the patient touches the first peg.	- El cronómetro se pondrá en marcha en el momento que la persona evaluada toque la primera clavija.
	- The stop watch is stopped when the patient places the last peg in the container.	- El cronómetro se parará cuando la persona evaluada deposite la última clavija en el recipiente.
**Instalación** **(Mathiowetz et al., 1985 [14])**	- A square board with 9 holes: - holes are spaced 3.2 cm (1.25 inches) apart - each hole is 1.3 cm (0.5 inches) deep	- Tablero cuadrado (de madera o plástico) con 9 agujeros: - Distancia entre los agujeros: 3.2 cm [14,30] o a 5.0 cm [31]. - Profundidad de los agujeros: 1.3 cm
	- 9 wooden pegs should be 0.64 cm (0.25 inches) in diameter and 3.2 cm (1.25 inches) long.	- Tamaño de las clavijas: 0.64 cm de diámetro y 3.2 cm de largo.
	- A container that is constructed from 0.7 cm (0.25 inches) of plywood, sides are attached (13 cm × 13 cm) using nails and glue.	- Recipiente para las clavijas: caja cuadrada (100 × 100 × 10 mm) separada del tablero) o un hueco redondo y cóncavo al final del tablero [15].
	- The peg board should have a mechanism to decrease slippage. Self-adhesive bathtub appliqués were used in the study.	- El tablero debe presentar un mecanismo para reducir la posibilidad de deslizamiento. Se pueden utilizar pegatinas autoadhesivas antideslizantes.
**Instrucciones** **(Mathiowtz et al., 1985 [14])**	- The instructions should be provided while the activity is demonstrated.	- Las instrucciones deben proporcionarse mientras se realiza la demostración de la actividad.
	- The pegboard should be placed in front of the patient, with the container holding the pegs on the side of the dominant hand.	- El tablero debe situarse en frente de la persona evaluada, con el recipiente que contiene las clavijas en el lado de la mano dominante
	- The patient’s dominant arm is tested first.	- Se debe evaluar primero el brazo dominante.
	- Instruct the patient to: “Pick up the pegs one at a time, using your right (or left) hand only and put them into the holes in any order until the holes are all filled. Then remove the pegs one at a time and return them to the container. Stabilize the peg board with your left (or right) hand. This is a practice test. See how fast you can put all the pegs in and take them out again. Are you ready? Go!”	- Indique a la persona evaluada las siguientes instrucciones: “Coja las clavijas de una en una, usando sólo su mano derecha (o izquierda) y métalas en los agujeros en el orden que quiera hasta que estén todos llenos. A continuación, saque las clavijas de una en una y vuelva a dejarlas en el recipiente. Sujete el tablero con su mano izquierda (o derecha). Esto es un ensayo. Vamos a ver lo rápido que puede poner y sacar todas las clavijas. ¿Está preparado? ¡Empezamos!”
	- After the patient performs the practice trial, instruct the patient: “This will be the actual test. The instructions are the same. Work as quickly as you can. Are you ready? Go!” (Start the stop watch when the patient touches the first peg.) - While the patient is performing the test say “Faster” - When the patient places the last peg on the board, instruct the patient “Out again…faster.” - Stop the stop watch when the last peg hits the container.	- Después de que la persona evaluada realice el ensayo, indíquele: “Esta será la prueba real. Las instrucciones son las mismas. Hágalo lo más rápido que pueda. ¿Está preparado? ¡Empezamos!” (Ponga en marcha el cronómetro cuando el sujeto evaluado toque la primera clavija). - Mientras la persona evaluada esté realizando la prueba diga “Más rápido”. - Cuando la persona evaluada coloque la última clavija en el tablero, indíquele “Ahora sáquelas …. Más rápido”. - Pare el cronómetro cuando la persona evaluada deje la última clavija en el recipiente.
	- Place the container on the opposite side of the pegboard and repeat the instructions with the non-dominant hand.	- Coloque el recipiente en el lado opuesto del tablero y repita el procedimiento con la mano no dominante.

**Table 3 healthcare-12-00550-t003:** Distribution of NHPT scores according to sex, educational level, and employment status (*n* = 40).

Sociodemographic Variables (*n*)			NHPT-E SCORE		
	Hand	Mean (SD)	Median (IR)	Minimum	Maximum
Total (40)	D	16.55 (2.52)	16.32 (14.83–17.58)	12.35	23.50
	ND	17.43 (2.56)	16.62 (15.74–18.78)	13.86	23.54
Age					
20–39 (20)	D	16.08 (2.5)	15.40 (14.71–17.27)	12.35	23.50
	ND	16.72 (2.08)	16.01 (15.63–17.39)	14.13	22.67
40–59 (20)	D	17.01 (2.52)	16.84 (15.53–18.08)	13.43	22.35
	ND	18.15 (2.84)	17.14 (16.20–20.37)	13.86	23.54
Sex					
Women (24)	D	16.57 (2.19)	16.42 (15.48–17.62)	12.35	21.83
	ND	17.57 (2.30)	16.75 (15.89–18.78)	15.14	22.67
Men (16)	D	16.52 (3.03)	15.05 (14.48–17.36)	13.43	23.50
	ND	17.22 (2.98)	16.27 (15.14–18.91)	13.86	23.54
Educational level					
Primary Education (23)	D	17.06 (2.09)	16.97 (16.06–17.67)	13.90	21.83
	ND	18.35 (2.38)	17.39 (16.70–20.35)	13.86	21.66
Secondary Education (11)	D	17.17 (2.30)	16.42 (15.08–18.98)	13.43	23.50
	ND	18.35 (2.39)	15.75 (15.43–16.14)	13.95	22.67
Higher Education (16)	D	15.70 (2.40)	15.12 (13.98–17.01)	12.35	22.35
	ND	17.28 (2.549	16.59 (15.82–17.70)	14.13	23.54
Employment status					
Active (35)	D	16.08 (2.04)	16.09 (14.65–17.27)	12.35	21.83
	ND	17.54 (2.38)	16.80 (15.88–19.16)	13.86	22.67
Unemployed (5)	D	19.77 (3.42)	19.96 (18.12–22.35)	14.92	23.50
	ND	16.70 (3.90)	15.46 (14.39–15.94)	14.13	23.54

D: dominant; *n*: number of participants; ND: non-dominant; SD: standard deviation; IR: interquartile range.

## Data Availability

Data are contained within the article.

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
