# Peer review of "Translation, Cross-Cultural Adaptation, and Feasibility of the NHPT-E of Manual Dexterity for the Spanish Population"

_healthcare, 2024, doi:10.3390/healthcare12050550_

Round 1

Reviewer 1 Report (Previous Reviewer 1)

Comments and Suggestions for Authors

We would like to congratulate the authors for their rigorous work in reviewing this manuscript. We would only have liked to have seen a revision of the references used, for more recent references.

Author Response

Dear reviewer, 
The authors of this paper sincerely appreciate your suggestions. We take your comments into account to improve this work. About your suggestion" We would only have liked to see a revision of the references used, for more recent references", the authors have updated references 2,4,5,9 and 10 cited in the manuscript.

Reviewer 2 Report (Previous Reviewer 2)

Comments and Suggestions for Authors

Thank you for the opportunity to review this paper again. This is a timely and important issue to explore that i believ will merit publication.

Please kindly consider the following points and revise them.

- please remove the dot at the end of the title

- line 36: please remove the ellipsis in brackets (e.g. eating, dressing, writing, ...)

- 2.2.1, 2.2.2, 2.2.3. remove the colon sign after the subsection names, please.

Author Response

Dear Reviewer,

The authors of this work really appreciate your suggestions. We take your comments into account for improving this paper.

Question:  please remove the dot at the end of the title, line 36: please remove the ellipsis in brackets (e.g. eating, dressing, writing, ...) and 2.2.1, 2.2.2, 2.2.3. remove the colon sign after the subsection names, please.

RR: The authors have modified the relevant formatting details.

This manuscript is a resubmission of an earlier submission. The following is a list of the peer review reports and author responses from that submission.

Round 1

Reviewer 1 Report

Comments and Suggestions for Authors

We would like to thank you for the opportunity to review your manuscript. The area chosen is relevant to health professionals. Here are our suggestions for improving the report. The merit of your work is not in question; however, some aspects need clarification or reformulation.

We suggest the alteration of the manuscript title to: Translation, cross-cultural adaptation, and feasibility of the Nine Hole Peg 2 Test of manual dexterity for the Spanish population. This was in fact what was performed.

How were the two translated versions consensualized? This part of the translation process is unclear. Please clarify. What was the role of the experts? Conceptual equivalence? Content validity? What were the characteristics (criteria to be considered as such) of the experts? How were they identified and recruited?

Figure 1 refers to the translation process. The introduction of the term "linguistics" raises questions about the context in which it is used.

Do not repeat in the text information that is already on the tables. Either remove the tables or the information in the text.

It is not clear why authors decided to perform correlational statistics in a cross-cultural adaptation process. The calculation of the content validity index would be understandable, but it is not clear the purpose of these calculations in a methodological study. What were the guidelines used to guide statistical analysis? What was the purpose of the pilot study? Was it for validation purposes? For feasibility? Authors must clarify the use of correlational statistics in the pilot study. It looks like this would be a secondary outcome from data collection: analyzing correlations between dexterity and sociodemographic characteristics.

Ln. 193: 95% of the sample or 95% of the participants?

Ln 213: viability or feasibility?

Discussion should be reviewed according to methods and results review.

Ln. 266: internal validity or reliability?

Methodological guidance for the translation, cross-cultural adaptation and feasibility is not clear. Beaton and colleagues’ (2007) guidance (Recommendations for the Cross-Cultural Adaptation of the DASH & QuickDASH Outcome Measures) were developed for self-administered questionnaires, therefore, the number of participants for the pilot study is that these authors recommend and not suited for this study. It seems that the pilot study was used to determine feasibility. The use of guidelines to translate and cross-culturally adaptation for self-reported outcomes may not be the more suitable, considering that the authors sought to translate and cross-culturally adapt the Nine Hole Peg 2 Test of manual dexterity, whose instructions are used and administered exclusively by health professionals.  

Some of the issues mentioned in the Limitation section should be acknowledge in the Methods section. The limitations are a statement of the possible bias that require special attention when interpreting the results. The strategies used to reduce risk of bias must be acknowledge in the Methods section.

More than 50% of the references used are over 5 years. Authors are strongly advised to review, using more up-to-date evidence.

Author Response

Dear Reviewer 1, the authors of this work really appreciate your suggestions. We take into account for improving this paper and we also try to answer the questions or explained the different issues in more detail.

Reviewer 2 Report

Comments and Suggestions for Authors

Thank you for the opportunity to review this paper. This is a timely and important issue to explore that i believ will merit publication. The aims of this study was the translation and culturally adapt the original version of the NHPT. It is clear paper and the  conclusions are sound. There is, however, few issues that must be resolved before the study can be accepted for publication.

Please kindly consider the following points and revise them.

- Lines 43-44:” There are different standardized tests that evaluate aspects of fine motor skills, such as speed and precision of movement, grasp and release, writing skills and hand posture.” – Please add references.

- Figure 1 – poor quality of graphics, figure is difficult to read.

- Lines 254-259: „Therefore, by way of conclusion, this study demonstrates that the adapted version of the NHPT for the Spanish population; the NHPT-E is presented as an instrument: 1) simple, clear and easy to understand, in accordance with the proposal of the original version, 2) it maintains the same response format, and 3) the changes made have improved the understanding of the test for participants and the administration and correction process for evaluators” – this should be a separate section CONCLUSIONS

- 4.1. Clinical Applicability and Future Research – this should be in the text after the conclusions

- Limitations – authors should include this in the text at the end of the discussion before conclusions

- In the whole manuscript: research articles usually do not use the word "we" and regularly use passive verbs. There are punctuation errors.

Author Response

Dear Reviewer 2, the authors of this work really appreciate your suggestions. We take into account for improving this paper and we also try to answer the questions or explained the different issues in more detail.

Round 2

Reviewer 1 Report

Comments and Suggestions for Authors

We acknowledge the author's effort to meet the suggestions made. However, the authors failed to meet the major concerns presented in the first round of this review process.

Authors use different references for translation and adaptation guidelines (at least one of these references is not acknowledged in the reference list), some of which are contradictory, which does nothing to help the coherence and consistency of the report. 

The study is relevant and of interest for practice, especially for those who use this device, but the authors have failed to adequately report the process. 

At this point, I believe it would be best for the authors, with the necessary peace of mind and time, to reflect on the manuscript and how they report the results of their study and submit it later.